# Compounds Identified from Marine Mangrove Plant *(Avicennia alba)* as Potential Antiviral Drug Candidates against WDSV, an In-Silico Approach

**DOI:** 10.3390/md19050253

**Published:** 2021-04-28

**Authors:** Mohammed Othman Aljahdali, Mohammad Habibur Rahman Molla, Foysal Ahammad

**Affiliations:** Department of Biological Sciences, Faculty of Science, King Abdulaziz University, P.O. Box 80203, Jeddah 21598, Saudi Arabia; mrahmanmolla@stu.kau.edu.sa (M.H.R.M.); fahammad@stu.kau.edu.sa (F.A.)

**Keywords:** In-silico drug design, virtual screening, *Avicennia alba*, gag polyprotein, homology modeling, ADMET, molecular dynamics simulation

## Abstract

Walleye dermal sarcoma virus (WDSV) is a type of retrovirus, which affects most of the adult walleye fishes during the spawning time. The virus causes multiple epithelial tumors on the fish’s skin and fins that are liable for more than 50% of the mortality rate of fish around the world. Till now, no effective antiviral drug or vaccine candidates have been developed that can block the progression of the disease caused by the pathogen. It was found that the 582-amino-acid (aa) residues long internal structural gag polyprotein of the virus plays an important role in virus budding and virion maturation outside of the cell. Inhibition of the protein can block the budding and virion maturation process and can be developed as an antiviral drug candidate against the virus. Therefore, the study aimed to identify potential natural antiviral drug candidates from the tropical mangrove marine plant *Avicennia alba*, which will be able to block the budding and virion maturation process by inhibiting the activity of the gag protein of the virus. Initially, a homology modeling approach was applied to identify the 3D structure, followed by refinement and validation of the protein. The refined protein structures were then utilized for molecular docking simulation. Eleven phytochemical compounds have been isolated from the marine plant and docked against the virus gag polyprotein. Three compounds, namely Friedlein (CID244297), Phytosterols (CID12303662), and 1-Triacontanol (CID68972) have been selected based on their docking score −8.5 kcal/mol, −8.0 kcal/mol and −7.9 kcal/mol, respectively, and were evaluated through ADME (Absorption, Distribution, Metabolism and Excretion), and toxicity properties. Finally, molecular dynamics (MD) simulation was applied to confirm the binding stability of the protein-ligands complex structure. The ADME and toxicity analysis reveal the efficacy and non-toxic properties of the compounds, where MD simulation confirmed the binding stability of the selected three compounds with the targeted protein. This computational study revealed the virtuous value of the selected three compounds against the targeted gag polyprotein and will be effective and promising antiviral candidates against the pathogen in a significant and worthwhile manner. Although in vitro and in vivo study is required for further evaluation of the compounds against the targeted protein.

## 1. Introduction

Due to the growing global demand of fish protein, a number of aquaculture industries have been developed to fulfil the demand where capture fisheries are continuously shrinking around the world [1]. Nonetheless, emerging threats originating from different viruses have great influences on fish farms and capture fisheries growth and actual production [2]. Emerging viruses are affecting both farmed and wild fishes, which have tremendous impact on economical aspect around the world [3]. Walleye dermal sarcoma virus (WDSV) is one of them, and crucially impact fishes during their spawning time. It was first isolated from walleye fish (Stizostedion vitreum) in the USA [4]. This retrovirus usually occurs at 4 °C in the winter and infection of the epidermis of fish leads to different types of neoplastic disease. This neoplastic disease is transformed to an invasive tumor within 12–16 weeks and causes the highest mortality of fishes in many countries. [4]. There is no obvious treatment option to control the spread of the virus due to lack of effective therapeutics drug.

The structural proteins of a virus play an important role in viral encapsulation, genome assemble, protection from genome degradation, maturation and releasing the mature virus to make re-infection with the host cellular protein through different mechanism [5,6]. The envelope structure of WDSV and their structural protein carry essential components, which helps viral maturation resulting in releasing of the mature virus and infectivity to the host. The structural and enzymatic proteins of the virus are ex-pressed as gag and gag-pro-pol (and in some cases gag-pro) polyprotein, respectively. The gag polyprotein assembles to form an immature viral particle, which is later matured through different proteolytic cleavages that induces significant structural changes of the protein products. Initially, the viral enzyme protease (PR) domain cleaves the gag-pro-pol into different precursors during or after assembly and budding. The PR then bind with the trans on gag and gag-pro-pol that help to produce mature proteins of the virus. The gag gene is encoded with the viral enzyme protease (PR) which synthesized as a unique large precursor polypeptide and fuse to gag [7]. The gag gene of the WDSV virus also encodes different mature proteins of unknown function that regulate the transcription and RNA processing of the virus. The process is responsible for induction of WDSV and inhibition of the gag polyprotein of the virus can block the transcription process and generate effective antiviral against the virus.

Compounds originated from marine plants have been found to show antiviral and drug like properties for many diseases. The marine plants and their constituents play a vital role in the drug design study as well as animal and human life savior [8]. Different plant herbal extracts have led to identifying single molecule as drug candidates and are being used in different biochemical and biomedical sectors [9]. *Avicennia alba* is a member of true mangrove flowering plants of the genus Avicennia and family Acanthaceae. This genus contains 11 species and several subspecies with many active biological constituents in this plant. The family members of the Acanthaceae have been utilized as traditional and alternative medicine for many years. Previously, leaf extract of *Avicennia marina,* a member of Avicennia, has shown activity against replication of herpes simplex virus type-1 (HSV-1) and vaccine strain of polio virus [10]. On the other hand, several phytochemicals such as querctin, lutolin, baicalein and kaempferol originated from plant extract have been also used as antiviral candidates agents against HIV and Dengue virus for many years [11,12]. Therefore, the extract and subsequent compounds originated from *Avicennia alba* can be utilized as replication inhibitors for different virus as well as WDSV.

Finding viable phytochemicals is a critical objective for the drug discovery [13]. The identification of targeted therapeutics to the fish industries require selection of small bioactive molecules by investigating the action of a molecules with the targeted protein [14]. In-silico drug design techniques are an easy and low time consuming process that can predict the liable compound against a specific drug targets [15]. The ideal compounds against a specific target can be selected based on molecular docking based scoring function and interaction can be documented through the analysis of the different docking poses [16]. The ADMET properties that indicate the efficacy and toxicity of compounds can be easily predicted through computer aided methods, where MD simulation confirm the stability of a drug candidate to the targeted protein. However, it has been difficult to address new drugs, therefore there is a need to progress strong investigation for inventing bioactive compounds by targeting novel protein classes. Therefore, we have planned to utilize an in-silico approach to investigate the new drug candidates against the WSDV virus targeting the gag polyprotein.

## 2. Results

### 2.1. Protein 3D Structure, Refinement, and Validation

The best 3D protein structure was downloaded from the I-TASSER server. The serve generated top five models of the gag protein and based on C-score further evaluation has been performed. The lowest C-score was −4.73. After refinement, the protein model-5 was chosen, where the 3D refine score was 37759.1, GDT-HA score was 0.9747, RMSD value was 0.347, MolProbity 3.473. The RAMPAGE and ProSA-web servers were also used to validate the refine gag protein. Before refinement, the Ramachandran plot analysis of the gag protein found 81.1%, 11.89% and 6.7% residues in the favorable allowed and disallowed regions accordingly, where refined gag protein model showed 91.65, 4.97, and 3.38%, residues in the favorable, allowed and disallowed regions, respectively (Figure 1A). Likewise, the crude model of the gag protein has a Z-score value of −4.14 and after refinement the value has improved to −4.75 (Figure 1B).

### 2.2. Phytochemical and Protein Preparation

The compounds of marine plant *Avicennia alba* has been retrieved from the popular database IMPPAT. Eleven phytochemical compounds of *Avicennia alba* plant have been retrieved and saved in a 2D (SDF) file format. The compounds have been prepared and optimized during the ligand preparation steps and converted into pdbqt file format for further evaluation. The protein was optimized and prepared for molecular docking process by using AutoDock tool and saved in pdbqt file format for docking.

### 2.3. Active Site Identification and Receptor Grid Generation

Active site (AS) of an enzyme is formed by combining different AA residues in a specific region that are able to make a temporary bond with the substrate known as the binding site. AS of a protein allows to bind with a chemical substrate resulting catalyzed reaction. It also helps to stabilize the intermediates of the reaction, where the binding site is a position of a protein or nucleic acid that can recognize ligand and can make a strong binding interaction with the protein. The study first identified AS of the gag poly peptide from CASTpi server then the combined binding position of the active site was retrieved (Figure 2). Analysis of the active pocket of the protein helped to retrieve the binding site residue of the protein (Figure 2). Active site pocket analysis revealed binding site position at MET1, PRO8, LEU13, LYS14, ASP49, GLU59, LEU63, ILE108, THR232, THR233, PRO235, ILE240, GLN250, TYR256, ARG280, GLN313, ARG324, MET402, MET416, THR426 residual positions that have been represented in ball shape with different colors red, green and orange as shown in Figure 2. The binding sites identified by the server have been utilized to generate a receptor grid during molecular docking simulation process and a grid box dimension X = 91.39, Y = 48.45 and Z = 53.14 in angstrom (Å)

### 2.4. Molecular Docking Simulation

A possible and accurate drug-like small molecular candidate was not only determined through the molecular docking, but also selected by macromolecule interaction and can also be identified from the method. The docking process can identify both protein and ligands forming best intermolecular frameworks for adjusting with host. Phytochemical compounds and desired proteins were selected, and the best intermolecular interaction determined through the molecular docking study. Eleven phytochemical compounds were utilized for molecular docking process by using PyRx tools AutoDock Vina wizard. The binding affinity showed a distributed range from −3.2 and −8.5 after molecular docking of phytochemicals compound. The top fifty percentage phytochemical compounds have been chosen from the 11 compounds based on the capacity of top binding affinity. The best three compounds, namely Friedlein (CID 244297), Phytosterols (CID12303662), and 1-Triacontanol (CID68972) have been selected based on their docking score −8.5 kcal/mol, −8.0 kcal/mol and −7.9 kcal/mol and further evaluated through different screening methods. The best three compounds selected based on molecular docking score are listed in Table 1 and docking scores for all compounds are listed in Appendix A. Additionally, the docking score for the selected three compounds was validated through the re-docking process. The single binding poses of the selected three compounds have been retrieved and re-docked again at the same binding site of the compounds. In this, case the upper and lower RMSD, having the low value, were chosen and the binding affinities found for the compounds were approximately same as previous score. The compounds CID244297, CID12303662, and CID68972 showed a docking score −8.4 kcal/mol, −8.1 kcal/mol and −8.1 kcal/mol, respectively, and has RMSD value of zero.

### 2.5. Protein-Ligand Interaction Analysis

Gag polyprotein with the highest binding score producing compounds has been selected and retrieved to observe the interaction between them. The interaction formed between the selected three ligands with the desired protein was observed by using the BIOVIA Discovery Studio Visualizer tool. For the compound CID244297 it was found to form several hydrogen and hydrophobic bonds with the desire gag polyprotein. The hydrogen bonds found to formed at ALA364 position, where the hydrophobic bonds from at the position ILE240, ILE342, PRO369, and PHE365 position is shown in Figure 3 and the bond types are listed in Table 2.

In the case of compound CID12303662 it has been observed to form several hydrophobic bonds in the ILE240, ILE342, ALA364, ALA364, and PRO369 residual position. Another two conventional hydrogen bonds were found to form at the position of VAL346 and PHE365 AA position as shown in Figure 4 and are listed in the Table 2.

For the compound CID68972 it was found to form the highest binding interaction with the desire protein. The compound has formed three hydrogen bonding interactions at GLY418 and PRO420 positions. A total of fourteen hydrophobic bonds were also found to form with the desired compounds, including alkyl bonds in the LYS56, LYS368, LEU415, MET416, PRO420, ILE427, MET416, LEU63 residual positions as depicted in Figure 5 and listed in Table 2.

### 2.6. ADME Analysis

The analysis of pharmakon (drug) and kinetikos (movement), combinedly known pharmacokinetics (PK) properties analysis, was an indispensable part during drug design and development process. However, it is mainly analyzed at the ADME properties and includes physiochemical properties such as lipophilicity, water solubility, pharmacokinetics, medication likeness and medicinal chemistry and provided the possible hypothesis for selecting the best drug candidates. Before undergoing drugs into the preclinical studies, the analysis of pharmacophore properties can determine the xenobiotic regulation features of a compound. The SwissADME server have been used to determine the pharmacophore properties from the selected three drug-like compounds. The drug-like compounds have a lipophilicity feature which dissolve in fats, oils, and nonpolar solvent. Finally, pharmacophore properties have shown the suitable result with compound thus can be used as an effective and druggable in the study. The pharmacokinetics properties found for the selected three compounds have been listed in the Table 3.

### 2.7. Toxicity Prediction

To know the adverse effect of a specific compound on animal, plant and human or the outside of the environment, it is important to measure the toxicity of compounds. Therefore, toxicity prediction has become a first and crucial step for selecting any compound as a drug. The toxicity analysis requires different animal models, which is time consuming and expensive process. On the other hand, the computer-based toxicity test is currently requiring no animal model and is not time consuming and expensive and can be used instead of the conventional method. In-silico toxicity test, we have chosen the selected three compounds through the popular web server admetSAR 2.0 and ProToxII. The server has predicted hepatotoxicity, carcinogenicity, immunotoxicity, mutagenicity, and cytotoxicity by using rat as a target model which given in Table 4.

### 2.8. MD Simulation Analysis

Molecular dynamic simulation has been run for exploring the binding stability of protein-ligand docking complexes. The molecular dynamic simulation also documented evidence based on intermolecular interaction through the orientation time. However, intermolecular interaction confirmation we got from the molecular dynamic simulation that has observed by the 100 ns simulation run between the protein and ligand complex structure. The analysis of RMSD, RMSF and protein ligand interaction mapping has been examined through the simulation trajectories.

### 2.9. RMSD Analysis

The RMSD can measure the average atomic distance of protein and help to characterize the protein [17]. The RMSD value can calculate the difference between observed value and estimated value and a value change between 1–3 Å or 0.1–0.3 nm is reasonably acceptable [18,19]. In our study, the RMSD value for the protein structure (Cα) residues and ligand fit protein were documented from the 100 ns simulation trajectory. The acceptable variation of ligand with protein was calculated from 100 ns interval during RMSD analysis and depicted in Figure 6. However, the RMSD calculated from Cα atoms of gag polyprotein and ligand fit protein showed the considerable fluctuation for all the three compounds. The three compound CID244297, CID12230662, CID68972 have found the optimum fluctuation >3.0 Å. The fluctuation for all the three ligands was high before 40 ns simulation time and after 40 ns the fluctuation for all compounds was reduced and showed a state of equilibration resulting in stability of the compound to the binding site of protein.

### 2.10. RMSF Analysis

RMSF provided the information about the local change of protein along with protein chain. The local changes of protein as well as protein chain determined from the RMSF value can help to characterize the protein. Therefore, the RMSF of the protein in complex with the selected phytochemical compounds were calculated for the PubChem CID244297(gray), CID12230662 (Yellow) and CID68972 (red) shown in Figure 7. The highest peak point of the fluctuation has observer between 150 and 155 residual point. The all complex structure has also been fluctuated from 500 to 550 residues due to the location of N-terminal domain. Overall, the fluctuation for all the compounds found optimal compared to the RMSF of the gag protein. Therefore, the compounds will be able to maintains a stable interaction without changing the structure of the protein.

### 2.11. Protein-Ligand Contact Mapping

The gag protein interactions with the phytochemicals were monitored throughout the analysis of 100 ns MD simulation trajectory. These interactions of the compound’s CID244297, CID12230662 and CID68972 have been categorized in different types, like hydrogen bonds, hydrophobic, ionic and water bridge bonds and summarized in the stacked bar charts as shown in the Figure 7. The substantial interaction with catalytic residues shows the significant interaction for all the selected three compounds. The screened compounds have shown the considerable intermolecular contact to the molecular dynamic simulation interval. It was found that the active residues not only keep contact with intermolecular but also with selected ligands as well as calculated their density for all of three compounds. The depicted figure shows that the dark area orange color indicates ligand in the specific frame with residues of protein (Figure 8). The active pocket of gag protein was derived through the screening of natural compounds, which was established by the analysis of molecular dynamic trajectories. Hence, the screened natural compounds CID244297, CID12230662, and CID68972 will be stable with the gag protein in real life experiments.

## 3. Discussion

In the modern drug design, computer aided drug design (CADD) brings a new medicinal era by providing cost effective processes, reducing time and minimize labor recruitment that make it reasonable in the process of drug discovery. It is an indispensable part as well as a tool for drug design [20]. Therefore, scientists and researchers have been affording the context of biological and synthetic research by the accelerating of CADD. As a result, selecting drug candidate shows that the highest biological efficacy identifying through the molecular docking, ADMET, and molecular dynamic simulation process [21]. A disease can be minimized by understanding the mechanism of the disease, identifying the disease associated protein and ligand binding approach to the protein. Understanding the process can reveal compounds that can interact and inhibit a specific protein resulting blocking the mechanism of disease. CADD help to identify exact target molecule based on their behavior and binding mode of the ligand. On the other hand, molecular docking clarifies the predominate binding modes within a ligand and protein and the MD simulation disclose the complex mechanism of protein ligand interaction. Thus, small molecule candidates can be identified as drug candidates against a specific disease.

The study utilized a compressive drug design approach that helped to screen eleven natural phytochemicals compounds by targeting the gag polyprotein to fight against the WDSV. The best three compounds have been chosen from the eleven compounds which have the highest binding affinity found from the molecular docking score. The higher binding has been documented to the compound CID244297, CID12230662, CID689702 with binding score −8.5, 8.0 and −7.9 kcal/mol, respectively. The ADME approaches have been utilized to emphasize the metabolite kinetics of small molecular candidates in the body. This was also the time-consuming evaluation process and conventional drug design processes required part of the body of animals or others. The ADME is mainly acting on the drug related to the pharmacokinetic properties. The PK parameters should be optimized before drug design process because it needed to pass standard clinical trial for a promising drug candidate [22]. This property is not only affecting the small molecules permeability across the biological system but also includes molecular weight and polar surface topological area with it. (TPSA). The high molecular weight may cause permeability in the drug candidate and TPSA helps to increase the permeability of lower ones [22]. Lipophilicity is the ability of chemical compound dissolve in polar and non-polar solvents and is expressed by LogP. It not only referred to the inorganic logarithm but also coefficient of the target molecules through the aqueous phase partition. Therefore, it influences various drug molecule’s absorption within the body and suggested that lower absorption rate spontaneously corelated to higher logP. However, LogS shows the negative attitude and affect the solubility of the candidate molecules that are considered the lowest value. Hydrogen bonds affect the capacity of drug molecules crossing the bilayer of the membrane, which depends on the number of donors and acceptors of hydrogen bonds. The rotatable bond is overwhelming with oral bioavailability because of their high rational barrier. The PK properties have been implemented to all of compounds and required evaluation has performed that found good value for all of the three compounds.

Moreover, toxicity test has been designed to generated data concerning the adverse effect that can harm or damage an organism. It was documented that about 20% of drug development failed due to the positive range of toxicity. The test of toxicity is indispensable part before drug experiment that is much expensive, and time consume of apply to animal trial [23]. Thus, in-silico alternative methods have been chosen before drug development because of no need to require animal trials and is even time-consuming and reasonable. In this study, three phytochemical compounds identified have shown low toxic effect and have optimize PK properties.

MD simulation helped to determine the physical movements of a compounds within a desired macromolecular environment, which has become an important tool in the CADD process [24]. The MD simulation helps to analyze the stability of the desired drug candidate with a targeted macromolecule. Therefore, MD simulation has been done in this study to observe the RMSD, RMSF, and ligand-protein interaction of the complex system, which found the optimum RMSD and RMSF value with good protein-ligand contact for all the three compounds. Therefore, the selected three compounds can be design further class of antiviral drug against the WDSV.

## 4. Materials and Methods

### 4.1. Homology Modeling

The WDSV structural gag polyprotein sequence (UniProtKB: O92815) has been retrieved through the UniProtKB (https://www.uniprot.org/uniprot/O92815) database accessed on 31 January 2021. To ensure or predict the three-dimensional (3D) structure of the desire protein the retrieved sequence was submitted into the popular online web portal on 31 January 2021 (https://zhanglab.dcmb.med.umich.edu/I-TASSER/), namely Iterative Threading Assembly Refinement (I-TASSER) [25]. The I-TASSER server created top five models of the protein structure and provided C-score, TM-score value and root-mean-square deviation (RMSD) of the protein structure. Top protein 3D structure was selected and downloaded as a PDB format considered on the C-score value. The higher value of C-score indicates that the protein model was high confidence from positive to negative numbers.

### 4.2. Model Refinement and Validation

Protein structure refinement has been done after submitting 3D structure into online web-based server on GalaxyRefine (http://galaxy.seoklab.org/cgi-bin/submit.cgi?type=REFINE) accessed on 7 February 2021. The RMSD value, energy score and quality of the final structure were identified from the GalaxyRefine server [26]. The maximum and minimum RMSD were determined and the average distance between atoms as well as energy score was used as the parameters to select the refined structure. PyMol v2.3.4 software is used to visualize the refined structure [17]. Ramachandran plot scoring function was used to validate the refine model. Furthermore, Ramachandran plot score (drug design) and z score value is lead to evaluate 3D structure that determine the standard deviation through the main value [27]. However, Z-score plot has documented through the popular server at ProSA-web (https://prosa.services.came.sbg.ac.at/prosa.php) accessed on 8 February 2021. The allow and disallow regions of amino acid are considering by Ramachandran plot that was studied through the Rampage server (https://warwick.ac.uk/fac/sci/moac/people/students/peter_cock/python/ramachandran/other/) accessed on 8 February 2021 [28].

### 4.3. Protein and Ligand Preparation

Prior to docking, refined and validated protein structures need to be prepared. Therefore, chosen 3D structure of gag protein was prepared through AutoDockTools (ADT). The gasteiger charge not only calculated from the protein but also merged and added non-polar and non-polar hydrogen. It also acts as a discarded metal ion and cofactors from its protein. Indian Medicinal Plants, Phytochemistry And Therapeutics (IMPPAT) database available at (https://cb.imsc.res.in/imppat/home) has been used for searching the desire plant compound and a total 11 compounds of marine plant *Avicennia alba* were downloaded from the database on 8 February 2021. Selected compounds retrieved from the database that has been set and minimized energy through the Universal Force Field (UFF) designed for individual ligand.

### 4.4. Protein Active Site Identification and Receptor Grid Generation

The refined protein structure of gag protein has been submitted in CASTp 3.0 web server available at http://sts.bioe.uic.edu/ accessed on 8 February 2021. The server identified different active pockets of the protein and based on the pocket surface area (SA); the first active pocket of the protein has been selected. From the active pocket the binding site residues of the protein has been identified and visualized through BIOVA Discovery Studio Visualizer Tool 16.1.0. The binding sites identified from the server were utilized for receptor grid generation during molecular docking simulation in PyRx tools.

### 4.5. Molecular Docking Simulation

Virtual screening of selected compounds was performed through molecular docking process that was carried out by PyRx software. Many potential drugs have been identified against several diseases through the virtual screening software [29]. It is provided Lamarckian genetic algorithm (LGA) as a scoring function included both AutoDock and AutoDock Vina for docking. Molecular docking interaction were proceeding by PyRx tools AutoDock Vina in this study [30]. The BIOVA Discovery Studio Visualizer Tools 16.1.0.41 used for analyzing the complex binding poses [31].

### 4.6. ADME Analysis

The key criteria of compounds to develop a successful drug candidate is the ADME properties of a compound and should be evaluated. In-silico based molecules ADME predicted at the early stage of drug development can reduce the failure rate in clinical trial, because many drugs were not fit in the clinical trial and its demand before [32]. Therefore, the observation of ADME test is indispensable part of early-stage drug design process. The evacuation of any drug inside of the fish through the urine and feces, are at the final stage that directly affected by the ADME profile [33]. The internal impacts on the fish are on physiochemical properties, hydrophobicity, lipophilicity, gastrointestinal environment, blood brain barrier before execration. The ADME properties such as bioavailability and solubility profile, and GIT absorption were evaluating through the free accessible server at Swiss-ADME (http://www.swissadme.ch/) for the 11 selected compounds accessed on 10 February 2021 [22].

### 4.7. Toxicity Analysis

To measure the toxicity, in-silico computational techniques have been used for evaluating the safety profile of selected compounds. Therefore, it ensured that compounds have toxic effect which may be critically harmful for the human and animal. The qualitive and qualitative ways were used to evaluate and determine the toxicity profile, where the mutagenicity, carcinogenicity, LD50 value, immunotoxicity wereevaluated [34]. The ProTox-II (https://tox-new.charite.de/for) web server has been utilized to document the toxicity of selected compound during this study period on 10 February 2021.

### 4.8. Molecular Dynamic Simulation (MD)

Thermodynamic constancy of the receptor-ligand system was analyzed through the Desmond v3.6 Program [35]. This program supported automated simulation and free energy perturbation (FEP) calculation included with multiple temperatures as prediction of equation of state (EOS). Again, TIP3P water model and OPLS2005 force field was used to run MD simulation [36]. The simulation of the larger complex structure was determined by orthorhombic periodic boundary condition while the system charge and place randomly for stabilizing sodium ions and their system neutralize electrically. The buffer box calculation method was used with the box shape and size 10 Å. NPT ensemble along with periodic boundary condition were used for performing molecular dynamic simulation [37]. The simulation trajectory has been retrieved and analysis through simulation interaction diagram (SID) available at Schrödinger Release 2020-2.

## 5. Conclusions

The WDSV is responsible for the induction of a multifocal benign cutaneous tumor known as WDS that is found with very high frequency in some localities. The effective therapeutics against the virus have not been developed yet to minimize the frequency of the virus infection. Therefore, this study has been developed to search natural and effective antiviral candidates against the virus to hinder cutaneous tumor development. The study has utilized CADD approaches to develop molecules through homology modeling, molecular docking, ADMET and MD simulation methods, which identified three potential antiviral candidates namely Friedlein, Phytosterols, and 1-Triacontanol against the virus that will be able to reduce the economic losses in fisheries sector. Although further study is suggested (in vitro and in vivo) to evaluate the effectiveness of the compounds against the virus.

## Figures and Tables

**Figure 1 marinedrugs-19-00253-f001:**
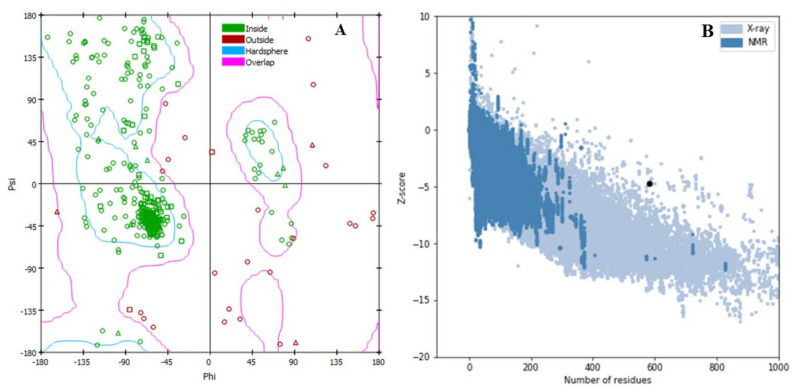
Validation of the 3D structure of the gag protein. (**A**) The Ramachandran plot statistics represent the most favorable, accepted, a disallowed region with a percentage of 91.65, 4.97, and 3.38%, respectively, and (**B**) the Z-score of refine gag protein −4.75.

**Figure 2 marinedrugs-19-00253-f002:**
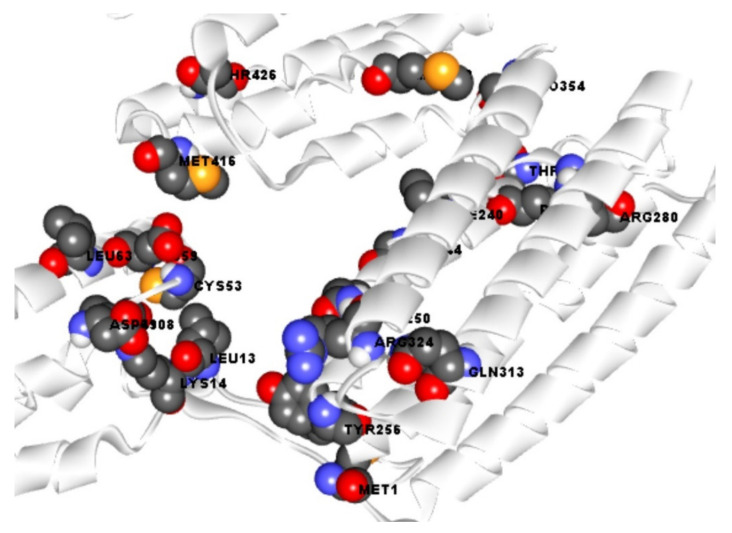
Showing the active site and correspondence binding site of gag polyprotein. Ball shapes with red, black, orange, and blue colors, respectively, with their binding site position of the gag polyprotein.

**Figure 3 marinedrugs-19-00253-f003:**
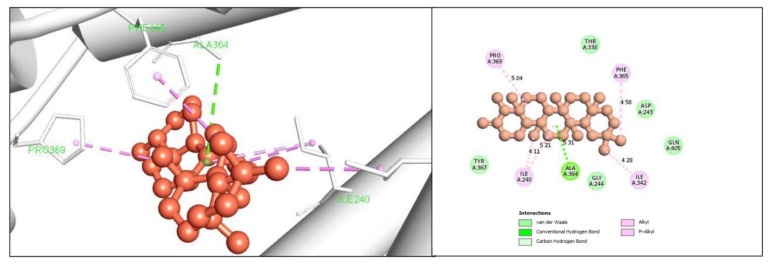
Shown the interaction between the compound CID244297 and gag polyprotein. Left side indicate 3D interaction and the right portion indicates 2D interaction of the protein -ligands complex.

**Figure 4 marinedrugs-19-00253-f004:**
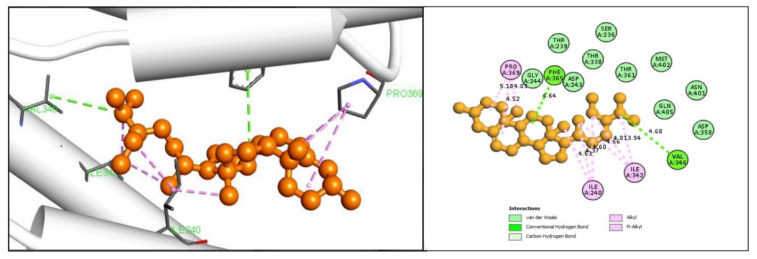
Shown the interaction between the compound CID12303662 and gag polyprotein. Left side indicate 3D interaction and the right portion indicates 2D interaction of the protein -ligands complex.

**Figure 5 marinedrugs-19-00253-f005:**
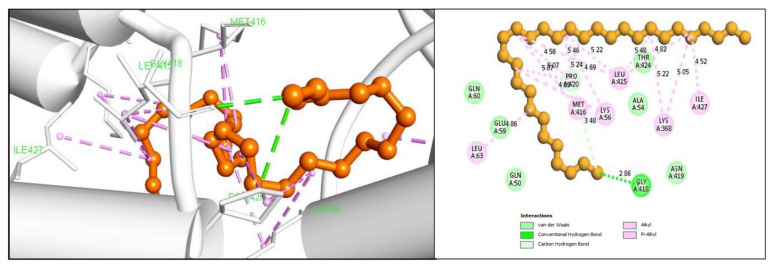
Shown the interaction between the compound CID68972 and gag polyprotein. Left side indicate 3D interaction and the right portion indicates 2D interaction of the protein -ligands complex.

**Figure 6 marinedrugs-19-00253-f006:**
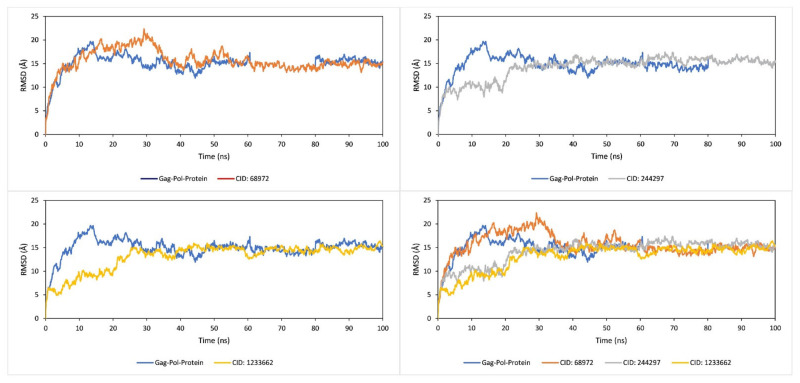
RMSD values extracted from the Cα atoms (blue curves) of gag polyprotein and natural compounds, where the compounds has shown in CID244297(orange), CID68972 (brown) and CID12230662 (yellow) with regards of 100 ns simulation time.

**Figure 7 marinedrugs-19-00253-f007:**
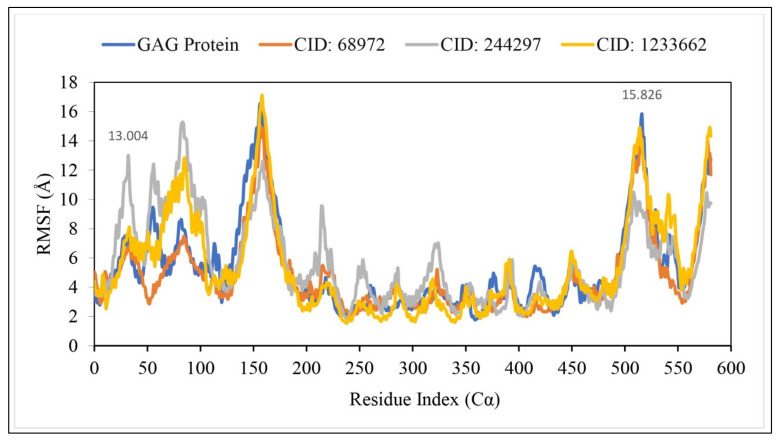
Showing the RMSF values extracted from the protein residue index Cα atoms of the complex structure viz. CID 244297(gray), CID12230662 (Yellow), CID68972 (red) and gag protein (blue) with respect to 100 ns simulation time.

**Figure 8 marinedrugs-19-00253-f008:**
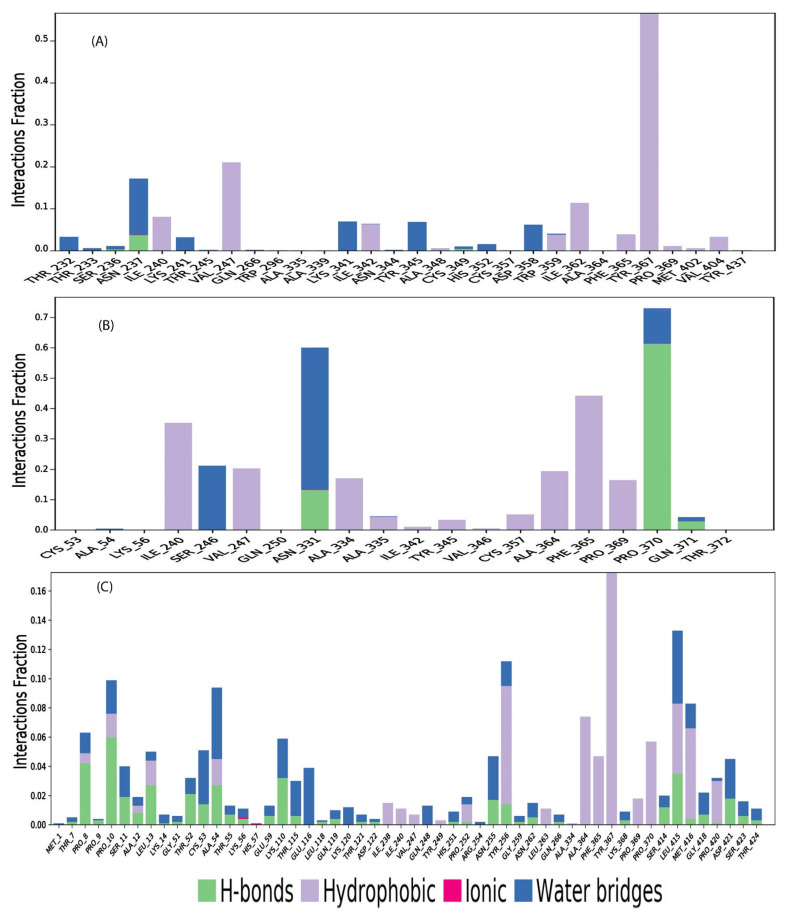
The stacked bar charts representing the contact mapping of gag protein with potential natural compounds, i.e., (**A**) CID244297, (**B**) CID12230662, and (**C**) CID68972 extracted from 100 ns simulations trajectory.

**Table 1 marinedrugs-19-00253-t001:** List of selected three compounds identified based on molecular docking score (kcal/mol) and their chemical name, formula, and correspondence PubChem CID.

PubChem CID	Chemical Name	Formula	Binding Affinity (kcal/mol)
CID244297	Friedlein	C_30_H_50_O	−8.5
CID12303662	Phytosterols	C_29_H_50_O	−8.0
CID68972	1-Triacontanol	C_30_H_62_O	−7.9

**Table 2 marinedrugs-19-00253-t002:** List of bonding interactions between selected four phytochemical with gag protein.

PubChem CID	Residue	Distance	Category	Type
CID244297	ALA364	5.30689	Hydrogen Bond	Conv-H-Bond
	ILE240	5.20546	Hydrophobic	Alkyl
	PRO369	5.03758	Hydrophobic	Alkyl
	ILE240	4.10592	Hydrophobic	Alkyl
	ILE342	4.27502	Hydrophobic	Alkyl
	PHE365	4.58263	Hydrophobic	Pi-Alkyl
CID6897	GLY418	2.86383	Hydrogen Bond	Conv-H-Bond
	PRO420	3.47712	Hydrogen Bond	C-H Bond
	LYS56	4.6933	Hydrophobic	Alkyl
	LYS56	4.99136	Hydrophobic	Alkyl
	LYS368	5.21921	Hydrophobic	Alkyl
	LYS368	5.05134	Hydrophobic	Alkyl
	LEU415	5.22063	Hydrophobic	Alkyl
	MET416	5.23747	Hydrophobic	Alkyl
	PRO420	4.55835	Hydrophobic	Alkyl
	PRO420	5.075	Hydrophobic	Alkyl
	LEU415	5.47725	Hydrophobic	Alkyl
	LEU415	4.82065	Hydrophobic	Alkyl
	ILE427	4.51538	Hydrophobic	Alkyl
	LEU415	5.46433	Hydrophobic	Alkyl
	MET416	5.07033	Hydrophobic	Alkyl
	LEU63	4.86074	Hydrophobic	Alkyl
CID12303662	VAL346	4.68285	Hydrogen Bond	Conv-H-Bond
	PHE365	4.64493	Hydrogen Bond	Conv-H-Bond
	PRO369	4.52399	Hydrophobic	Alkyl
	PRO369	5.17641	Hydrophobic	Alkyl
	ILE240	4.62631	Hydrophobic	Alkyl
	PRO369	4.03084	Hydrophobic	Alkyl
	ILE240	4.36608	Hydrophobic	Alkyl
	ILE342	4.65569	Hydrophobic	Alkyl
	ILE342	3.93691	Hydrophobic	Alkyl
	ILE240	4.59804	Hydrophobic	Alkyl
	ILE342	4.80551	Hydrophobic	Alkyl

**Table 3 marinedrugs-19-00253-t003:** List of pharmacokinetics includes ADME properties of the selected three compounds. The lists also present different physicochemical properties of the three compounds.

Properties		CID244297	CID12230662	CID68972
Physico-chemical attribute	MW (g/mol)	426.73 g/mol	414.72 g/mol	438.8 g/mol
	Heavy atoms	31	30	31
	Aromatic heavy atoms	0	1	0
	Rotatable bonds	0	6	28
	H-bond acceptors	1	1	1
	H-bond donors	0	1	4
Lipophilicity	Log Po/w	8.46	8.02	7.67
Water solubility	Log S (ESOL)	Soluble	Soluble	Soluble
Pharmacokinetics	GI absorption	Low	Low	Low
Drug-likeness	Lipinski	Yes	Yes	Yes
Medi. Chemistry	Synth. accessibility	Easy	Easy	Easy

Medi. Chemistry = Medicinal chemistry; Synth. Accessibility = Synthetic accessibility.

**Table 4 marinedrugs-19-00253-t004:** List of the drug-induced toxicity profile includes hepatotoxicity, carcinogenicity, immunotoxicity, mutagenicity, cytotoxicity of selected three compounds.

PubChem ID	Hepatotoxicity	Carcinogenicity	Immunotoxicity	Mutagenicity	Cytotoxicity
CID244297	Inactive	No	Inactive	Inactive	Inactive
CID12230662	Inactive	No	Light active	Inactive	Inactive
CID68972	Inactive	No	Inactive	Inactive	Inactive

## Data Availability

Not applicable.

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
