# Peer review of "Compounds Identified from Marine Mangrove Plant (Avicennia alba) as Potential Antiviral Drug Candidates against WDSV, an In-Silico Approach"

_marinedrugs, 2021, doi:10.3390/md19050253_

Round 1
Reviewer 1 Report
Author has study Walleye dermal sarcoma virus and used in-silico technique has been applied to identify potential natural antiviral drug candidates. The results indicated that Three compounds namely Friedlein (CID: 244297), Phytosterols (CID:12303662), and 1-Triacontanol(CID: 68972) have been selected based on their docking score -8.5 kcal/mol, -8.0 kcal/mol and -7.9 kcal/mol, respectively and evaluated through ADME (Absorption, Distribution, Metabolism and Excretion), and Toxicity properties.. In my opinion paper need to major revision.
1. Title is too long, Abstract and Introduction are quite premature and shallow. Ideally, readers expect to have a very brief account of the aims, methods, key findings, and conclusions of a study from an abstract with a couple of sentences from each part.
- Author explore binding affinity of protien using docking method, also mention the which amino acid are key residue for this this study. Redocking not reflecting here. It will better to validate the docking model through redock approaches.
- MD trajectory is not stable author should explore in details about RMSF and RMSD analysis fluctuation in mail text.
- Figures quality are not good. Author should change figure in high resolution and considered only H-bond.
Reviewer 2 Report
The manuscript entitled “Compounds identified from marine mangrove plant (Avicennia alba) as potential antiviral drug candidates against WDSV, an in-silico phytochemical screening approach” by Aljahdali et al. the study conducted on identification of compounds against WDSV through an in-silico approach is well and appreciated.
Minor corrections
- Authors used only in-silico approach, but my suggestion would be they can simply conduct the clinical test under controlled condition, that would enhance the value of the manuscript and study would suggest for others can use same in-silico method.
- Introduction - It needs to mention and give details that why and how you selected marine mangrove plant for exploring antiviral drug and also need what was the hypothesize authors had before beginning of the experiment.
- How authors get the information of 11 compounds from Avicennia alba
- “The RMSD values 1–3 Å or 0.1–0.3 nm is reasonably acceptable” (line 224) strengthen the sentence with suitable reference.
- What are the targets/models used in the study for toxicity analysis?
- Provide the figure 8 with clear pixel
- Line 276 and 282- What is CAAD? Is it CADD?
Specific comments
- Line 14 & 18- Gag or gag protein?
- Line 20 – not “than” it is then
- Line 68- “Querctin, Lutolin, Baicalein” use small letter “querctin, lutolin, baicalein”
- Line 105 & 108 - Avecenia Alva need to be corrected as “Avicennia alba”
- Line 116- What is the meaning for “AS”
- Line 244 –It is not between 150 to 155 residual point, it should be “between 150 and 155 residual point”.
Overall, the manuscript is satisfactory and informative. I have given minor comments. Also, I would suggest thorough proofreading for rectifying grammatical and usage errors if any. I would recommend publication of this manuscript after addressing minor changes.
Round 2
Reviewer 1 Report
We have assessed the findings presented here and referred to the previous literature. In the context of the data presented in the current paper improve to our previous expectations with regards to originality and we are therefore able to consider it for publication as present form.